# Semantic Matching Complements Saliency in Visual Search

## Abstract

Searching for a target in natural scenes can be guided by the semantic associations between the target and the scene, as well as the target-orthogonal properties of the scene. To estimate the semantic contributions to search, we analyze a database of human eye movements during visual search in naturalistic images, and construct an image-computable framework based on the CLIP model that exclusively leverages the match between the linguistic and the visual representations of the target and the scene, respectively. While this semantic matching model alone could explain a considerable portion of search behavior, weighting the model with saliency-based models could achieve a better prediction. These results suggest that our overt attention during search is constrained by not only the task at hand but also the task-orthogonal properties of the visual world.

**Keywords:** Visual search, Saliency model, Semantic matching

## 1. Introduction

In daily life, we often need to search for a semantically defined target, such as when asked to shop for "something to drink made out of fruit" in a grocery store. During the search, two conceptually distinct driving forces – target semantics and scene saliency – may guide our attention. On the one hand, the sensory attributes of the objects that are semantically related to the desired target (*e.g.*, oranges) may guide the search. On the other hand, the salient regions that are conspicuous to the sensory system (e.g., promotion banners) can also attract attention (Itti et al., 1998; Wolfe and Gray, 2007).

To determine the extent to which humans utilize the target semantics compared to scene saliency during visual search, we construct a computational model for human eye fixations, which estimates the target semantics through language-image associations. By weighting this semantic matching model with saliency-based models, we reveal the relative contributions of semantics and saliency to human search behavior.

## 2. Methods

### 2.1. Human visual search data

To inspect human visual search behavior in naturalistic scenes, we used the COCO-Search18 dataset (Yang et al., 2020; Chen et al., 2021), a laboratory-based database from 10 subjects. Each block of the search trials started with a target designation display with object exemplars, and the subjects were asked to identify whether the target was present or absent in a $54° \times 35°$ display scene, which remained visible until the response. Their eye movements were tracked using EyeLink 1000. MS-COCO images (Lin et al., 2014) were used for the search images and 18 categories were used for the targets. There were $2,489$ unique available tasks (pairs of a scene and a target) in the train and validation splits and 612 unique tasks in the test split, for each of the target-present and target-absent conditions.

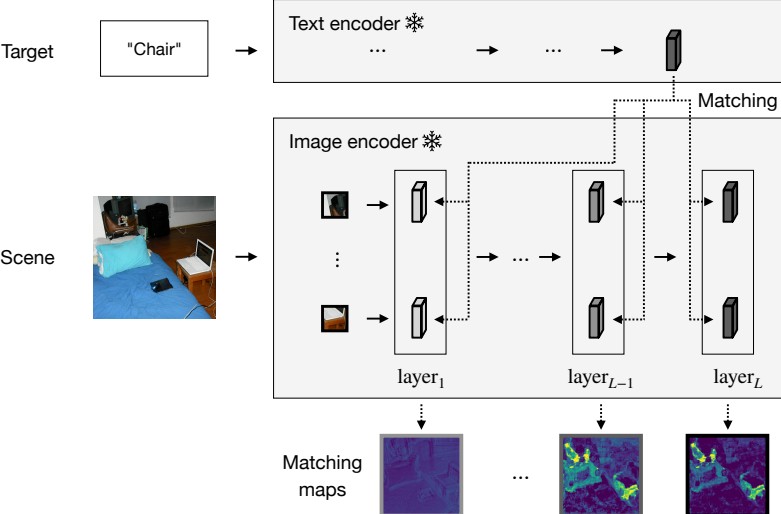

Figure 1: Semantic matching model. Matching maps are generated by matching the ultimate representation of the target input ("chair") and each of the intermediate representations of the scene input. $\text{layer}_1, \cdots \text{layer}_L$ denote the different layers of the image encoder. Solid arrows denote the streams to process images and targets. Dashed arrows denote the streams to generate matching maps.

### 2.2. Semantic matching model

To determine how much human search behavior can be accounted for by semantic relevance to the search target, we developed the semantic matching model (SMM), a model based on the Contrastive Language-Image Pre-training (CLIP) model (Radford et al., 2021). CLIP is contrastively trained with pairs of naturalistic images and captions, to learn similar representations at the final layers of image and text encoders, while all the layers of each of the encoders are allowed to be updated.

In the SMM, the scene (encoded by a vision transformer) is compared with the target (encoded by a text transformer) to generate a map of target-scene similarities (Fig. 1). These similarities indicate how likely each spatial location of the image is to be the fixation point of an eye movement. We used the pre-trained checkpoint `clip-vit-large-patch14` available at the huggingface repository[1]. In this CLIP model backbone, the image encoder splits the image into $16 \times 16$ patches and passes them through 24 vision transformer layers, and the text encoder has 12 layers. For comparison, we considered a model with the same architecture as the SMM, but with random weights in the image encoder ("random SMM").

Semantic matching is achieved by comparing text embeddings to image patch embeddings at all spatial locations via query-based attention. To search for the relevant semantic matches in image patch embeddings, we define a query vector $\mathbf{q}$ that reflects the relative specificity of the target information compared to all the different targets. Formally, let $\mathbf{t}$ be the 768-dimensional target vector given by the text encoder, followed by linear projection

---

1. https://huggingface.co/openai/clip-vit-large-patch14

and L2-normalization. Target vectors were constructed from the text inputs starting with "a photo of a[n]", followed by the names of the target categories. For the search target $k$, the query vector we used is given as follows.

$$\mathbf{q}_k = \mathbf{t}_k - \bar{\mathbf{t}} \tag{1}$$

where the target prototype $\bar{\mathbf{t}} = \sum_{i=1}^{N_{\text{targ}}} \mathbf{t}_i / N_{\text{targ}}$ where $N_{\text{targ}}$ is the number of different targets.

To determine locations in the image with visual features that correspond to the target category, we computed a match between the target-specific query vector and the image patch embeddings. Let $\mathbf{s}_{ij}^{(l)}$ be the 1024-dimensional spatial embedding vector, given by the $l^{\text{th}}$-layer output at $(i, j)$ patch location from the image encoder. With the query vector $\mathbf{q}_k$, the match response $r$ in the layer $l$ at the $(i, j)$ location $(1 \leq i, j \leq 16)$ is given as follows.

$$r_{ijk}^{(l)} = \sigma \left( -\mathbf{q}_k^\top \mathbf{W} \mathbf{s}_{ij}^{(l)} \right) \tag{2}$$

where $\sigma(x) = 1/(1 + e^{-x})$ is a sigmoid function and $\mathbf{W}$ is the $(768 \times 1024)$-shaped final projection matrix of the CLIP image encoder. We negatively signed the match between the query vector and the image embeddings because we found that the relationship between the text token and the image tokens was overall negative (see Appendix A). These matching responses $r$, lying between 0 and 1, were used as predictive maps for evaluation. For the high-resolution visualization purpose in Fig. 1, we ran the model $14 \times 14$ times independently with sub-patch level displacements.

### 2.3. Saliency models

We used the saliency-based models to characterize the task-orthogonal properties in scenes that guide human search behavior. Following Kümmerer, Wallis, and Bethge (2015), we define saliency as the properties of the scene features that attract fixations under the free viewing condition, going beyond the conservative definition as bottom-up low-level image conspicuity (Itti et al., 1998). For the main results, we used the DeepGaze-IIE model (Linardos et al., 2021) as a saliency estimator, with the model weights publicly available in the repository[2], along with the center bias log density computed from MIT1003 dataset (`centerbias_mit1003.npy`). Additionally, as an alternative saliency estimator, we used the human consistency under the free-viewing context where no targets are assigned, computed from the COCO-FreeView dataset (Chen et al., 2022; Yang et al., 2023). As a baseline, we used the Itti-Koch saliency model (Itti et al., 1998).

### 2.4. Evaluation of behavior prediction accuracy

To quantify how well a model predicts human fixations during visual search, we used a few different standard benchmark metrics (Kümmerer et al., 2018). We used the shuffled AUC (sAUC), the AUC from the uniform (uAUC), and the normalized scanpath saliency (NSS) for the metrics. For Figs. 2–3, we evaluated model performance by merging the train and

---

2. https://github.com/matthias-k/DeepGaze

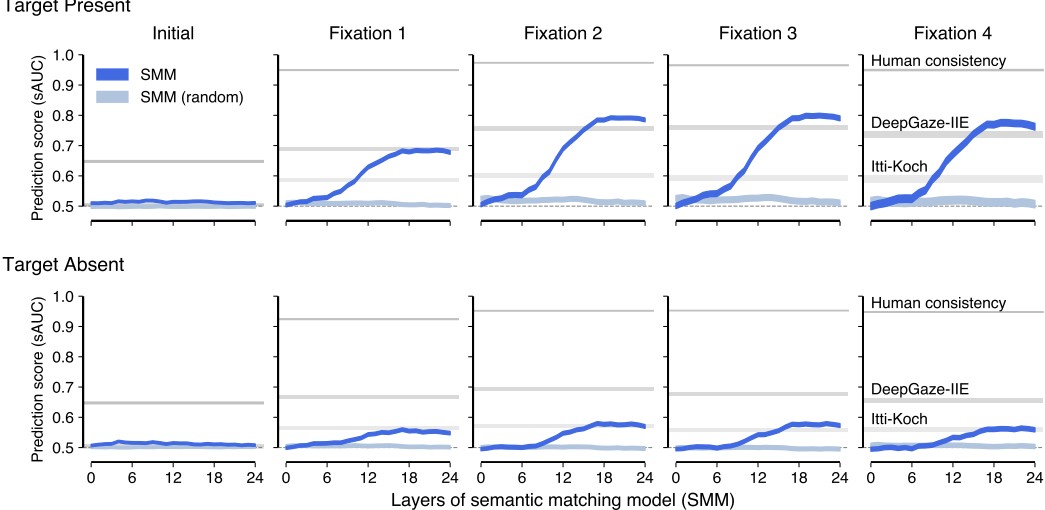

Figure 2: Semantic matching model explains human fixations during search. Each panel corresponds to the different fixation orders, when the target was present (upper) or absent (lower). Initial fixation was fixation forced to be at the center. For each panel, the blue curve is the average prediction score evaluated for each of the model layers. sAUC metric was used for model evaluation, with a theoretical chance level of 0.5. Colored shades denote 95% CIs.

validation splits of the COCO-Search18 dataset, as we have not used any of the data for training or fine-tuning the models that we consider here. When evaluating the models based on the optimal weights obtained from the train and validation splits in Tables 1, S1–S2, we used the test split. For order-specific evaluation, the scores were evaluated for each fixation order from the initial to the fourth fixation. For aggregate evaluation, we used the average scores across all the fixations after the initial fixations up to the search termination.

To factor out the center bias (*i.e.*, the image-independent distribution of fixations) in evaluation, we used sAUC in the main demonstrations (Figs. 2–3), as among AUC-type metrics, sAUC penalizes models with center bias (Kümmerer et al., 2015) whereas uAUC penalizes models without center bias. sAUC and uAUC quantify how a model distinguishes fixations (signals) from non-fixations (noises), by integrating a receiver operating characteristic curve for the model-predicted fixations versus false alarms for different criterion levels $\theta$ ranging among the possible values of map responses (here, the real line) as follows.

$$\text{AUC} = \int_{-\infty}^{\infty} \left(1 - \mathcal{F}_S(\theta)\right) \mathcal{F}'_N(\theta) d\theta \tag{3}$$

where $\mathcal{F}_S$, $\mathcal{F}_N$ denote the empirical distribution functions for signal (fixations) and noise (non-fixations) examples, respectively. For both sAUC and uAUC, we used the distribution of the map responses evaluated at fixation positions for $\mathcal{F}_P$. For the sAUC, we used the distribution of the map responses evaluated at fixation positions obtained in the other tasks for $\mathcal{F}_N$. For the uAUC, we used the distribution of the map responses of a given task across

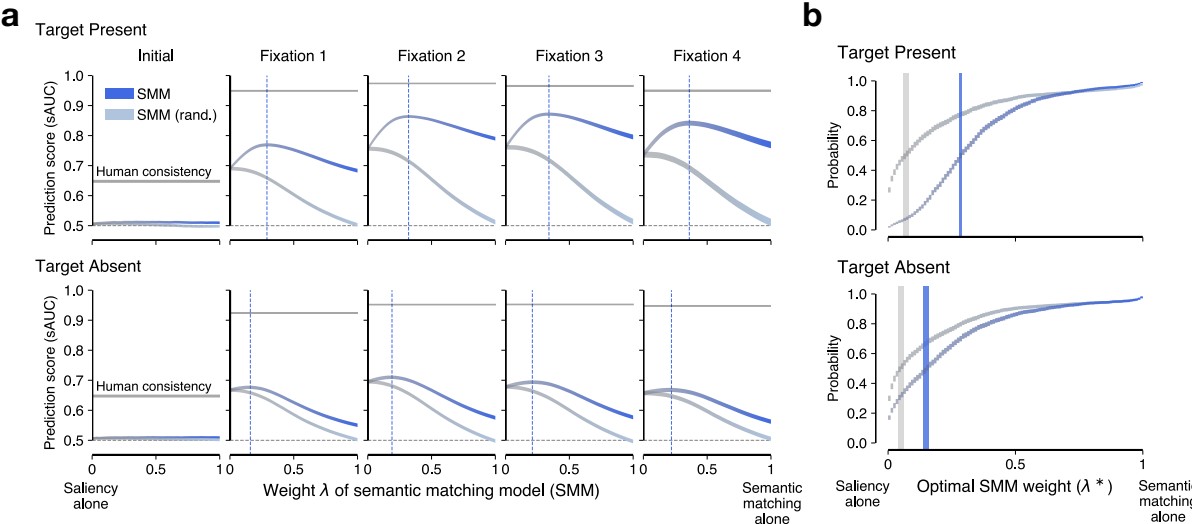

Figure 3: Balancing semantic matching with saliency better explains human fixations. **a** sAUC score predictions as a function of the balancing weight $\lambda$ of the semantic matching model against the saliency model (DeepGaze-IIE). Each panel corresponds to the different fixation orders, when the target was present (upper) or absent (lower). Dotted vertical lines denote the optimal weights that maximize the prediction score. **b** Empirical cumulative distributions of the optimal weights $\lambda^*$ that maximize the prediction scores. Vertical bars denote the bootstrapped median. Colored shades denote 95% CIs.

all the patch locations for $\mathcal{F}_N$. NSS is defined as the average standardized predictive map responses evaluated at fixations (Peters et al., 2005).

As a noise ceiling measure for each metric, we computed the human consistency by Gaussian kernel density estimates, with a standard deviation of 30 pixels (image size: 1680px × 1050px), approximately corresponding to the visual angle of 1°. We constructed the leave-one-subject-out density map by leaving out the subject to be evaluated and averaged the scores across different subjects. For a fair comparison across the aforementioned models, we averaged each map into 16 × 10 patches, in line with the output resolution of the SMM.

## 3. Results

### 3.1. Semantic matching predicts human fixations during search

We evaluated the accuracy of the SMM and found that it considerably predicted human fixations during visual search both in target-present and target-absent conditions. We measured the sAUC scores of the matching maps of the SMM as a function of fixation orders, compared to those of the random SMM (Fig. 2). When evaluated using all the fixations after the initial fixation, the SMM showed prediction performance significantly larger than the random SMM performance for most of the layers ($p < 0.05$ from 4th to 24th layers for target-present condition, $p < 0.05$ from 10th to 24th layers for target-absent condition, one-

tailed paired samples $t$-tests with the Bonferroni correction over 24 tests). This result was consistent when we used other metrics, NSS or uAUC, for scoring the performance (Fig. S2). These results suggest that a set of features learned from the training of the image and language alignment in the underlying CLIP model can be useful for predicting human fixations as in the SMM, both in target-present and target-absent cases.

Next, we assessed the performance of the SMM as a function of its layers and observed that there was a graded growth in prediction with increasing processing hierarchy. To capture the growth patterns, we considered a logistic function, $0.5 + (L-0.5)/(1+e^{-\kappa(x-x_0)})$, where $x$ denotes the layer index, and $L$, $\kappa$, $x_0$ the capacity, steepness and offset of the growth, respectively. Fitting the logistic function to the prediction scores across all the fixation orders after the initial fixation revealed a growth in the performance of the SMM as a function of its layers ($R^2 = 0.997$, $L = 0.759 \pm .011$ and $R^2 = 0.990$, $L = 0.560 \pm .009$ for the SMM in target-present and target-absent conditions, respectively, with bootstrap means and 95% CIs. $R^2 = -0.573$ and $R^2 = -0.128$ for the random SMM in target-present and target-absent conditions, respectively.).

Finally, we compared the SMM against the saliency models and found that the prediction growth capacity in the SMM exhibited significantly higher performance than the DeepGaze-IIE in the target-present condition (95% CI= $[0.025, 0.037]$, $p < 10^{-5}$, bootstrap test) but significantly lower performance in the target-absent condition (95% CI= $[-0.113, -0.104]$, $p < 10^{-5}$, bootstrap test). It is worthwhile to note that the SMM, without being explicitly trained to human eye movements, can outperform the DeepGaze-IIE when the targets are present. For the baseline comparison, while the SMM outperformed the Itti-Koch model in the target-present condition (95% CI= $[0.159, 0.173]$, $p < 10^{-5}$, bootstrap test), there was no significant difference between the SMM and the Itti-Koch model in the target-absent condition (95% CI= $[-0.003, 0.005]$, $p = 0.532$, bootstrap test).

Table 1: Fixation prediction test accuracy

| Models | Target Present | | | Target Absent | | |
|---|---|---|---|---|---|---|
| | sAUC | uAUC | NSS | sAUC | uAUC | NSS |
| $\lambda = 0$ | 0.723 | 0.833 | 1.174 | 0.663 | 0.813 | 1.112 |
| $\lambda = 1$ | 0.757 | 0.769 | 1.718 | 0.565 | 0.574 | 0.348 |
| $\lambda$ : selected | **0.831** | **0.893** | **1.954** | **0.675** | **0.818** | **1.147** |
| Human consistency | 0.926 | 0.947 | 4.828 | 0.814 | 0.881 | 2.431 |

### 3.2. Balancing semantic matching with saliency

An inspection of the relation between the SMM and the saliency model predictions showed that there was considerable variability in their predictions. There was insufficient evidence of the positive correlation between the prediction scores of the SMM and the DeepGaze-IIE, when normalized by the human consistency (Fig. S3, Pearson correlations $r = 0.003$, $p = 0.434$ for target-present conditions and $r = -0.040$, $p = 0.975$ for target-absent conditions, one-tailed bootstrap tests), suggesting that the models generate distinct predictions.

Motivated by this observation, we estimated how humans weight semantic contributions with saliency during visual search by considering a convex combination of the predictive maps, $\mathbf{r}^{\mathrm{SMM}}$ and $\mathbf{r}^{\mathrm{Saliency}}$, to balance their contributions to the prediction of human behavior.

$$\mathbf{z}_\lambda = \lambda \cdot \mathbf{z}^{\mathrm{SMM}} + (1 - \lambda) \cdot \mathbf{z}^{\mathrm{Saliency}} \tag{4}$$

where $0 \leq \lambda \leq 1$ and $\mathbf{z}$ is the standardization of predictive maps $\mathbf{r}$ (zero mean and unit variance). $\lambda = 0$ corresponds to the saliency model alone, whereas $\lambda = 1$ corresponds to the SMM alone. $\lambda$ lying between 0 and 1 indicates a weighted balancing of the saliency and the semantic contributions.

To determine whether the balancing helps to explain the human data, we evaluated the balanced predictive maps $\mathbf{z}_\lambda$ as a function of the weight $\lambda$. It turned out that there was a Goldilocks zone; overall, the intermediate weights of neither zero nor one best explained the human fixations, both in target-present and target-absent conditions. After the initial fixation, compared to the random SMM with monotonic decreases in the prediction scores, the SMM showed non-monotonic prediction curves with their maxima lying strictly between zero and one (Fig. 3a). Moreover, the optimal weights that maximize the predictive performance for given tasks were distributed toward the SMM (median weights $0.29 \pm 0.01$ and $0.15 \pm 0.01$ for target-present and target-absent conditions, 95% bootstrap CIs), compared to the random SMM (median weights $0.07 \pm 0.01$ and $0.05 \pm 0.01$ for target-present and target-absent conditions, 95% bootstrap CIs) (Fig. 3b).

We further verified that human fixations reflect a balance between semantic and saliency contributions by evaluating the generalizability of the balanced model with the test split of the dataset. Compared to when exclusively using the saliency model or the SMM, combining them with $\lambda$ selected across all the tasks in the train and validation splits led to better performance in the test split, both for the target-present and target-absent conditions in all the metrics (Table 1). We further replicated the generalizability of the balanced model using the DeepGaze-IIE model without the center bias as the saliency model (Fig. S4 and Table S1) and using the human consistency under the free-viewing context as the saliency model (Fig. S5 and Table S2).

Moreover, we show that the gain in the predictive performance from balancing is associated with the extent to which semantic matching and saliency models make different predictions. We found a positive correlation between the balancing gain from the saliency model and the discrepancy between the predictive maps of the SMM and the DeepGaze-IIE, measured as KL-divergence (Fig. S6, $r = 0.316$, $p < 10^{-58}$ for target-present conditions, $r = 0.315$, $p < 10^{-57}$ for target-absent conditions). This suggests that a distinct spatial distribution of predictive maps contributes to improving predictive performance.

## 4. Discussion

Here, we leveraged a new development in machine learning, the CLIP model, which allowed us to construct an ideal observer framework for semantically defined search problems to evaluate how well the model describes human search behavior and how it is weighted against saliency-based models. Under a specific model space, we found the optimal weights that balance the semantic and saliency contributions in explaining human eye movements.

A conundrum in the implementation of feature-based attention (Treue and Trujillo, 1999; Saenz et al., 2002; Zhang and Luck, 2009) is how the higher-level task-representing areas could provide specific enhancement for neurons in the downstream areas. A recent computational model suggested that the credit assignment onto the downstream neurons should be computed by back-propagation (Lindsay and Miller, 2018), but the back-propagation of gradients is still not generally accepted as biologically plausible.

By construction, our semantic matching model suggests an alternative scheme of feature-based attention. Specifically, the semantically relevant features could be selected by neural semantic matching, since there may be areas of the brain optimized to have aligned semantic and visual feature representations as in the CLIP model. These semantic features can exploit the spatial representations inherent to the architecture of the higher-level visual encoding stream. Such information can guide the classic spatial attention (Posner, 1980) in the downstream areas, capitalizing on the topographic organization readily available throughout the cortex (Wandell et al., 2005; Mackey et al., 2017), without any need for knowledge about the tuning profiles of nonspatial features in downstream neurons.

Our results show a nontrivial contribution of saliency to visual search, as the combination of the task-orthogonal saliency predictions with the semantic matching predictions better explains human fixations. Such contribution of saliency is intriguing, given that considering saliency is not directly required by the search task and in some cases can hurt the search efficiency. Perhaps, saliency as a default mode would give a rapidly available basis set of features that are generally useful across different types of tasks, including semantically defined search. Semantic matching may provide flexibility to specific task demands, enabling an adaptive behavior in a dynamic environment when complemented with saliency.

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

## Appendix A. Negative contrast in semantic matching model

In the SMM, the semantic match, defined as the dot product between the target $k$-specific query vector and the scene representations at $(i, j)$ location, $\mathbf{q}_k^\top \mathbf{W} \mathbf{s}_{ij}^{(l)}$, generates a contrast useful for characterizing the semantic similarities within a scene. Using the penultimate layer $l = 23$, we demonstrate that the dot product is formed as negatives. Using the images and annotations of the MS-COCO "train" split not used in the COCO-Search18 data, we characterized the representations of the image spatial patches that are within the target object locations, using the pre-trained CLIP `clip-vit-large-patch14` visual encoder and its randomized counterpart. Across all the search target categories used in the COCO-Search18 data, the semantic match dot products were overall distributed negatively in the pre-trained CLIP visual representations (Fig. S1). This gives an empirical grounding for the need to put the negative sign for the computation of semantic matching responses.

Table S1: Fixation prediction test accuracy (DeepGaze-IIE without center bias)

| Models | Target Present | | | Target Absent | | |
|---|---|---|---|---|---|---|
| | sAUC | uAUC | NSS | sAUC | uAUC | NSS |
| $\lambda = 0$ | 0.760 | 0.836 | 1.299 | 0.710 | 0.810 | 1.160 |
| $\lambda = 1$ | 0.757 | 0.769 | 1.718 | 0.565 | 0.574 | 0.348 |
| $\lambda$ : selected | **0.842** | **0.887** | **2.014** | **0.718** | **0.815** | **1.195** |
| Human consistency | 0.926 | 0.947 | 4.828 | 0.814 | 0.881 | 2.431 |

Table S2: Fixation prediction test accuracy (Human consistency under free-viewing)

| Models | Target Present | | | Target Absent | | |
|---|---|---|---|---|---|---|
| | sAUC | uAUC | NSS | sAUC | uAUC | NSS |
| $\lambda = 0$ | 0.736 | 0.831 | 1.463 | 0.669 | **0.802** | 1.294 |
| $\lambda = 1$ | 0.757 | 0.769 | 1.718 | 0.565 | 0.574 | 0.348 |
| $\lambda$ : selected | **0.823** | **0.869** | **2.152** | **0.676** | **0.802** | **1.327** |
| Human consistency | 0.926 | 0.947 | 4.828 | 0.814 | 0.881 | 2.431 |

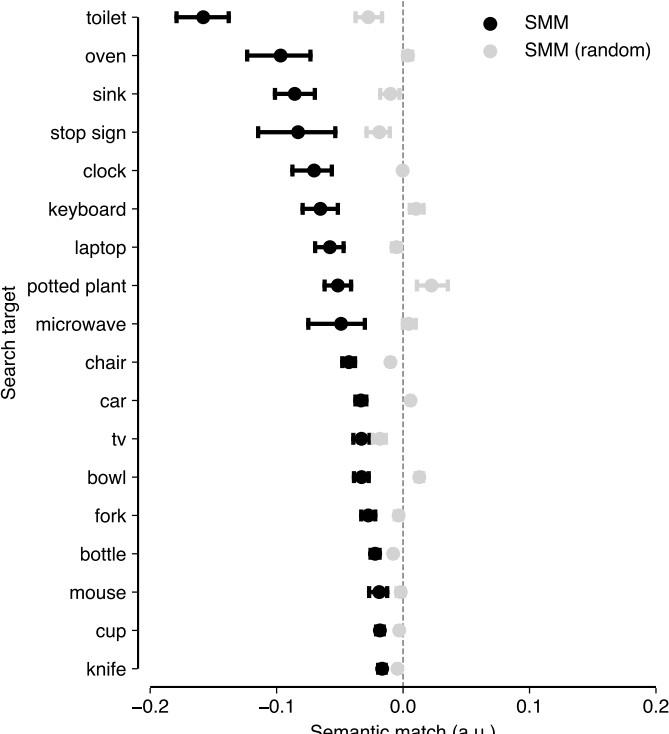

Figure S1: Negative match responses in the semantic matching model, evaluated between the representations of the search targets used in the data and the image patches that fall under the annotated object bounding box. For the images and the annotations, we used MS-COCO "train" split images not used in the COCO-Search18 data. Error bars denote 95% CIs.

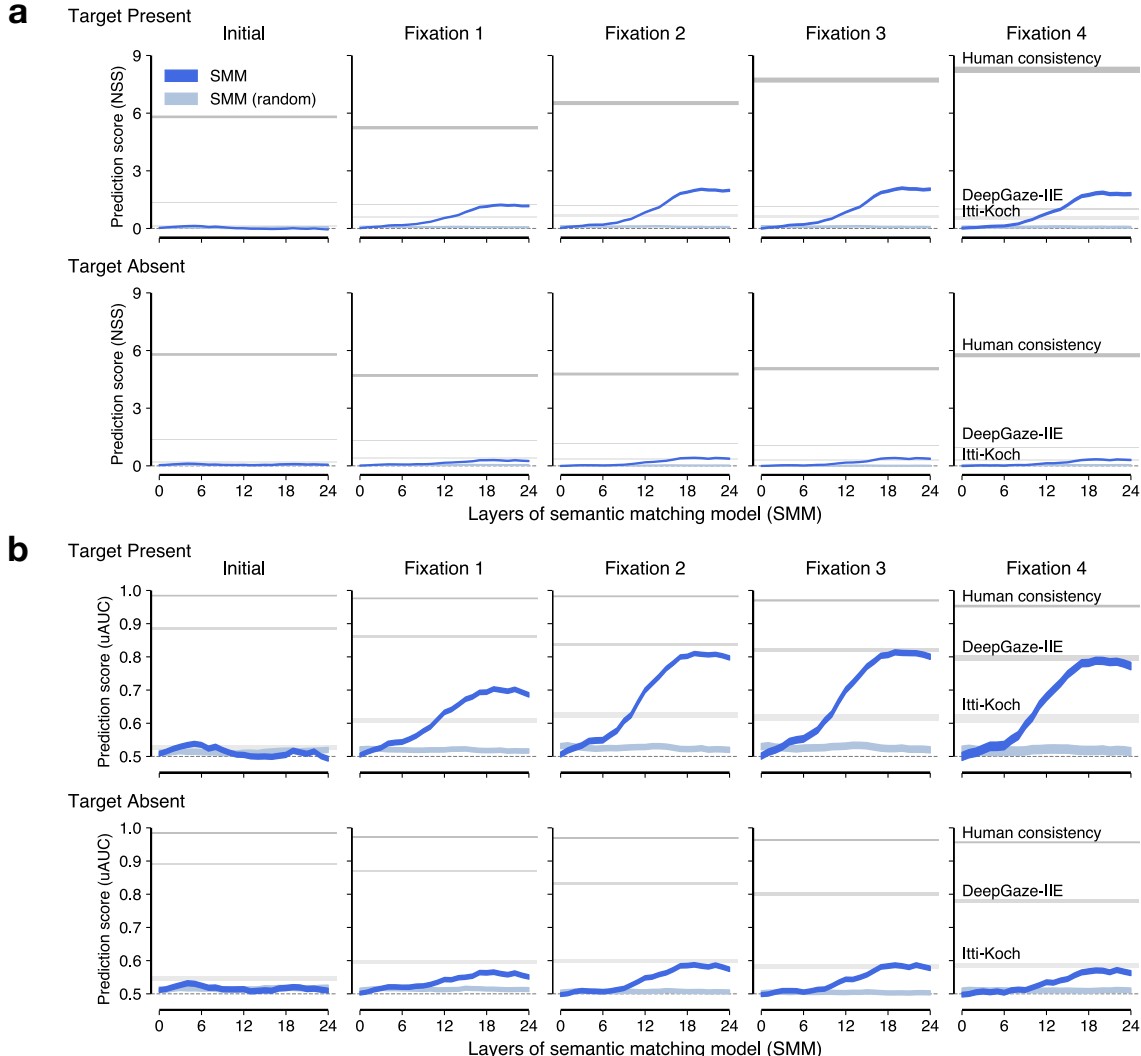

Figure S2: Same as the Fig. 2, but with NSS and uAUC metrics used for model evaluation.

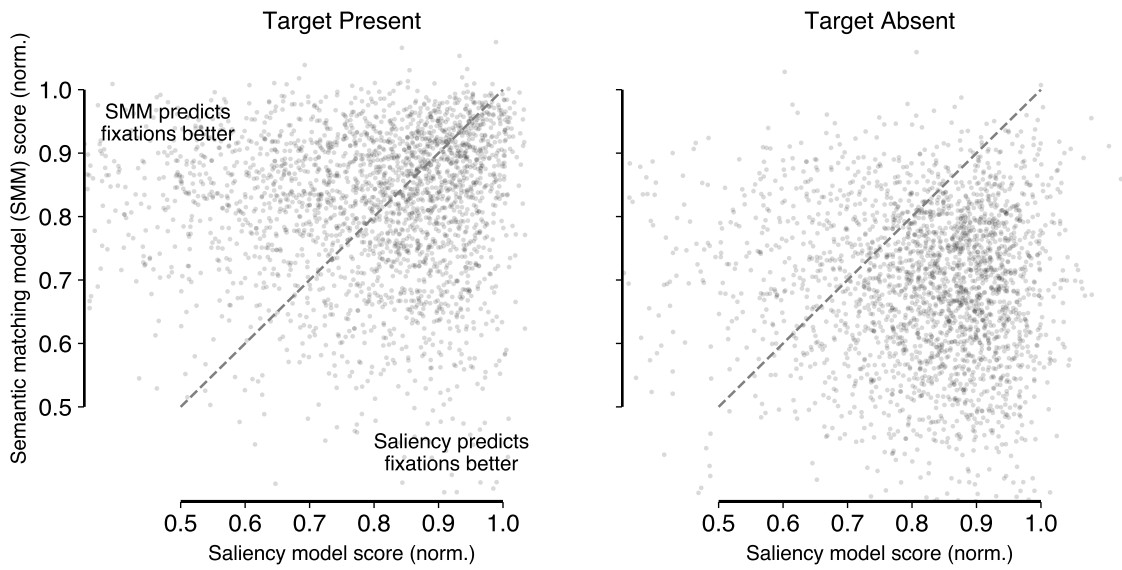

Figure S3: Distinct predictions by the semantic matching model and the saliency model (DeepGaze-IIE). Each dot denotes the scores for an individual task (image-target pair). Both scores were divided by the human consistency score. Pearson correlations $r = 0.003$ (target present) and $r = -0.040$ (target absent).

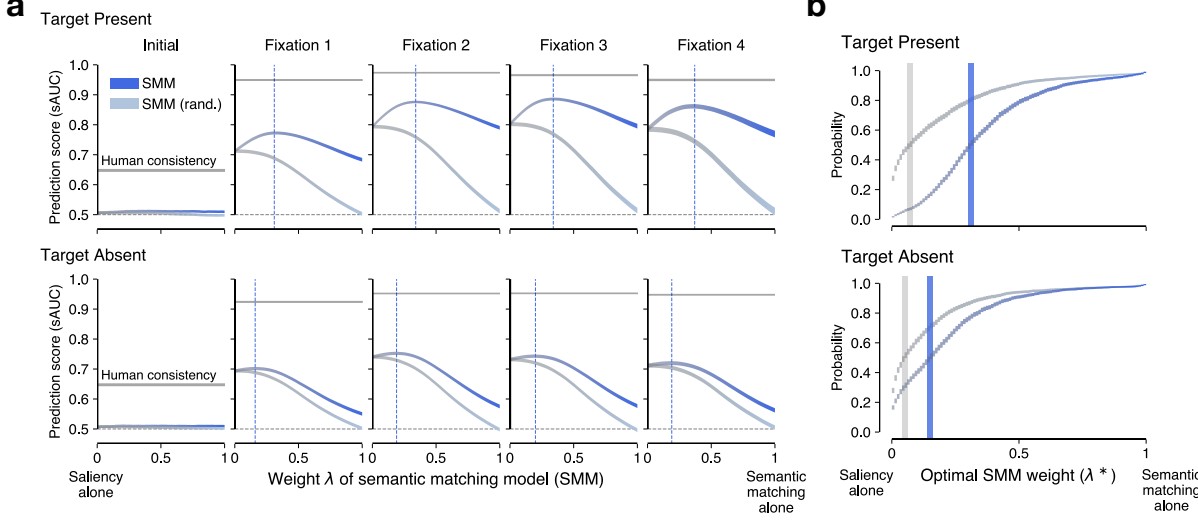

Figure S4: Same as the Fig. 3, but using the DeepGaze-IIE model without the center bias as the saliency model.

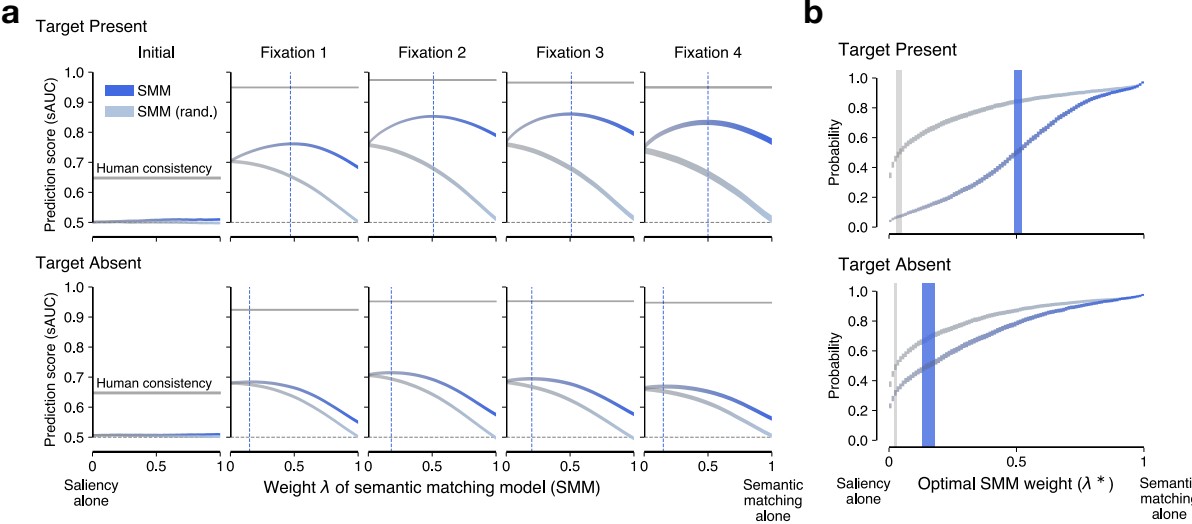

Figure S5: Same as the Fig. 3, but using the human consistency in free-viewing context as the saliency model.

## Appendix B. Predictive map discrepancies explain balancing gain

To quantify the dissimilarities in the predictive maps, we used the KL divergence (Le Meur and Baccino, 2013). To obtain KL divergence, we first converted the predictive maps into the probability maps using softmax function, $\mathbf{p}_{ij} = \dfrac{e^{\mathbf{r}_{ij}}}{\sum_{x,y} e^{\mathbf{r}_{xy}}}$.

$$D_{\mathrm{KL}}\left(\mathbf{p}^{\mathrm{SMM}}\|\mathbf{p}^{\mathrm{Saliency}}\right) = \sum \mathbf{p}_{ij}^{\mathrm{SMM}} \log\left(\mathbf{p}_{ij}^{\mathrm{SMM}}/\mathbf{p}_{ij}^{\mathrm{Saliency}}\right) \tag{S1}$$

To quantify how balancing improves human fixation prediction, we computed the normalized gain of the semantic matching model from the saliency model, by subtracting the sAUC score by the saliency model from the maximal sAUC score and dividing the quantity by the maximal sAUC score, for each of the tasks.

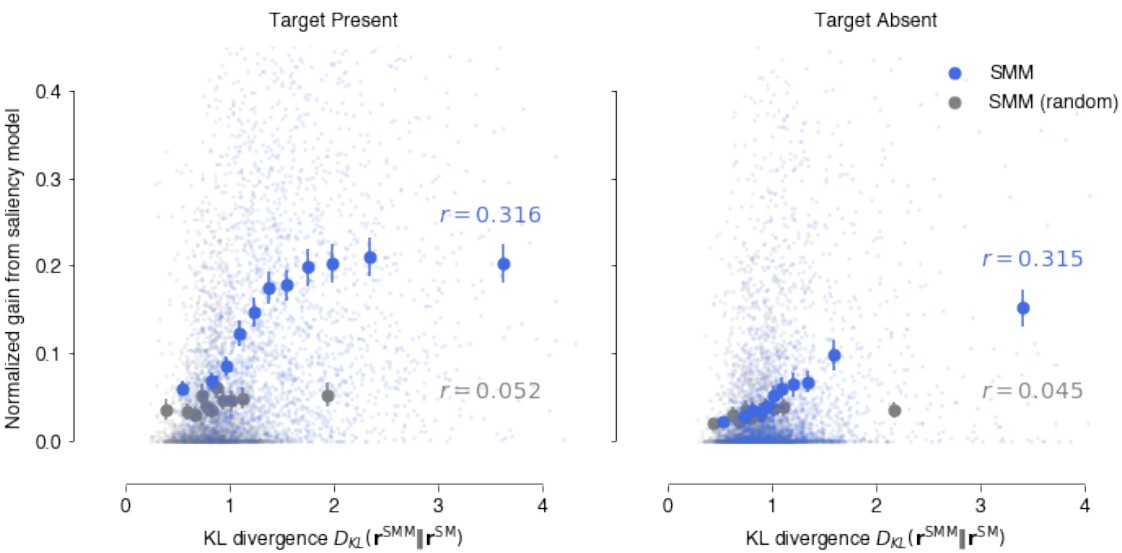

Figure S6: Predictive map discrepancies explain the balancing gain by the semantic matching model. Point ranges are constructed using equal-sized bins of KL divergences, with vertical bars denoting 95% confidence interval. Each data point corresponds to a unique image-task pair. $r$: Pearson correlation coefficient.

