# OpenReview forum: "Semantic Matching Complements Saliency in Visual Search"
_NeurIPS.cc/2023/Workshop/Gaze_Meets_ML — Submitted to Gaze Meets ML 2023_

### Official Review · Reviewer_kj1x · 2023-10-20
**An exploration considering semantic matching, visual search, and gaze**

**Rating:** 6
**Confidence:** 3

**Review:**

The authors consider the problem of semantic matching, visual search, and gaze association. The work is an interesting exploration of searching for a target in natural scenes and if it is guided by some high-level semantic association between visual embeddings and if there are influences of target-orthogonal properties of the scene. The experiment is constructed through an analysis of human eye movements
during visual search in naturalistic images, and a match between text and visual targets. The manuscripts exhaustively studies various aspects to determine influence of gaze in the matching process. They found that there exist optimal weights that balance the semantic and saliency contributions in explaining human eye movements. However, they express doubts about how semantically relevant features could be selected by neural semantic matching, since there may be areas of the brain optimized to have aligned semantic and visual feature representations. This is where assertions become a little hazy.

While their results show a nontrivial contribution of saliency to visual search, they also discovered that in some cases gaze semantics may hurt the search efficiency. It is unclear to the reader how findings from this work could be leveraged given this uncertainty. That said, it is certainly an interesting exploration that meanders through various considerations.

The algorithm is not clearly described, but their thinking and findings are well addressed. The reviewer is left unsatisfied with the outcome though intrigued by the process of the experiment itself. This is because there has been sufficient work in visual matching using text and the role of gaze in these matters isnt completely clear or fully leveraged. The authors could do more to explain the intuitive assessment of the. problem to improve the understanding of their contribution further - particularly as it relates to gaze and application of the findings.

However, the work is thought provoking and interesting; may be OK as a poster.

---

### Official Review · Reviewer_SX19 · 2023-10-22
**Review of Submission #23**

**Rating:** 4
**Confidence:** 2

**Review:**

SUMMARY : This paper constructs an image-computable model based on Contrastive Language Image Pretraining (CLIP) to study semantic saliency based visual search. Experiments are performed on a database of human eye movements during visual search in naturalistic images.

QUESTIONS/WEAKNESSES:

1. It is not very clear why the authors opted for the measure of saliency used in Eq. (2). They also state that "We negatively signed the match between the query vector and the image embeddings because we found that the relationship between the
text token and the image tokens was overall negative". Since the dot product is essentially computing measure of similarity, it is not clear whether this observation is an intrinsic property of the data or due to the form of the embedding vector constructed in Eq. (1) i.e. using mean subtraction. Is this pre-processing not at odds with the way CLIP was trained (i.e. on the vector dot product measure)? This central argument of the paper needs further clarification.

2. " Following Kummerer, Wallis, and Bethge (2015), we define saliency as the properties of the scene features that attract fixations under the free viewing condition, going beyond the conservative definition as bottom-up low-level image conspicuity" Please clarify what this means in simpler terms, it is quite hard to follow for readers not already familiar with these works. In general, providing more explanation for these baselines and the measures they propose for evaluation (at least in the appendix) would be really helpful.

3. How many cross validation splits were considered and what are the standard deviations for the measures in Tables 1 S1-S2 ? Since this dataset is comprised of only 10 subjects, it is unclear whether the findings are robust. Additionally, how was the $\lambda$ parameter in Eq. 4 set?

4. It is not obvious what the fixation order in Figs. 2-4 are. Please clarify. In general, please have the figures appear closer to where the main text content explaining the experiment is- this would improve the readability of the work.

---

### Official Review · Reviewer_5Ptr · 2023-10-24
**The authors describe the semantic matching model (SMM) framework to predict human fixations during visual search. Although the paper constructs an observer framework to describe human search behavior and human fixations, it leverages the CLIP framework and appears more like an application or enhancement to it.**

**Rating:** 7
**Confidence:** 3

**Review:**

The paper attempts to solve human search behavior and weights it against saliency based model, showing a balance between semantic and saliency contributions in explaining eye movements. A major highlight of the paper is combining the task orthogonal saliency predictions with semantic matching predictions. The authors have well demonstrated how both factors explain human fixations. Although well written and furnished with experiments, the paper comes across as an application/enhancement to the CLIP framework.
The paper is thorough and well written. The experimental setup and results have been discussed well. The authors might want to add in certain descriptions in the introductory section and make the furnished example more intuitive for the readers. It might also be worthwhile to mention relevant previous work/background section.

---

### Meta-Review · Area_Chair_ZH32 · 2023-10-26

**Recommendation:** Reject
**Confidence:** 4

**Metareview:**

The reviewers expressed concerns about the dot product similarity formulation in the paper. They felt that the language used in the paper needed more clarity and that the results and experimental methods needed further clarification. Additionally, some of the figures in the paper needed clarification as well.  While the study showed that saliency plays a nontrivial role in visual search, it also discovered that gaze semantics may reduce search efficiency in certain situations. The reviewers felt that this work needed clarification about how the findings of this study could be applied. Nonetheless, the study was an interesting exploration of various considerations.

Reviewers suggested that the authors could improve the understanding of their contribution by explaining the intuitive assessment of the problem, particularly as it relates to gaze and the application of their findings.

---

### Decision · Program_Chairs · 2023-10-26

Reject